# Local Materials as a Means of Improving Motivation to EFL Learning in Kazakhstan Universities

**Gulzhakhan Tazhitova, Dina Kurmanayeva \*, Kamaryash Kalkeeva, Jannat Sagimbayeva and Nurilya Kassymbekova**

Foreign Languages Department, L.N. Gumilyov Eurasian National University, Nur-Sutan 010000, Kazakhstan
\* Correspondence: dina_k68@mail.ru

**Abstract:** This study explores whether incorporating local materials into EFL instruction has a positive effect on students' motivation to learning a foreign language. The main purpose of this article is to show that if the content of foreign language teaching is systematized and constructed on local materials, then the content of foreign language teaching will contribute to an increase in motivation to learning a foreign language. This experimental method was used in order to identify the level of development of educational motivation in the course of using local materials. The results of the experiment showed that participants in the experimental group were more successful than those in the control group at the conclusion of the study, providing convincing evidence that the integration of local materials has a significant impact on increasing the motivation of EFL students to learning a foreign language.

**Keywords:** local materials; EFL; students; motivation; teaching; Kazakhstan

## 1. Introduction

One of the most pressing issues in education has always been creating positive motivation for studying. This topic has been the subject of numerous publications by teachers, psychologists, and other experts such as V.N.Myasishchev, M.M.Vasilieva, V.N.Kotkova, and others. Scholars consider motivation to be one of the most important variables in the success of the learning process; it represents a learner's level of interest in learning activities and serves to motivate them. According to V.N.Myasishchev, human activity is determined by 20–30% of his intelligence and 70–80% of his motivation. When a student's attitude is based on his or her own experience, activity setting, sentiments, emotions, desires, and interests, motivation is developed [1–3].

Despite the efforts of educators, the motivation to EFL learning remains low. In our opinion, factors such as uninteresting lessons, listening difficulties, reading difficulties, a large amount of homework, difficulty memorizing words and grammatical forms in the lesson, and unfamiliar material contribute to the decline in interest in EFL learning.

In this regard, methodologists/theorists and teachers/practitioners propose their own solutions to the problem of declining motivation to learn a foreign language, despite the fact that the subject is one of the priorities in the framework of modern education. Here are some particular suggestions for raising motivation in a foreign language class.

M.M.Vasilieva believes that in the classroom it is vital to create conditions that allow students to communicate solely in the target language. The author advises using quizzes, running Olympiads in a foreign language, and organizing debates with foreign students to accomplish this [2]. V.N.Kotkova believes it is a good idea to involve students in poetry recitation and dramatization in a foreign language [3].

One of the most important issues in improving the motivational level of students to EFL learning is the correct selection and adaptation of learning materials containing location-based information. According to Pauzan et al., the use of local content seeks to

provide a stock of knowledge, skills, and behaviors in order for students to have a consistent understanding of the status of the environment and community needs in accordance with the values/regulations that exist in the region to support regional and national development [4].

In the context of economic, social, and spiritual progress, one of the defining elements of the development of higher education is the orientation of students towards national identity through teaching local culture and traditions. This objective to implementation of the regionalization process cannot but affect the development and functioning of higher education.

Local materials present a comprehensive study of the culture and traditions of all the Kazakh people. The cultural component of education content consists of developing spiritual values, ideals, and character. Individuals who create a high national ideal will translate it into feelings, beliefs, and activities. Consequently, civic education can use their positions to create a sense of honors, duty, and patriotism.

In course of foreign language teaching, the use of local materials is presented as a secondary or additional teaching resource introduced by teacher into the foreign language content. Therefore, this should aim at instilling spiritual and ethnocultural values, ideals, and general social orientation in students.

The main purpose of this article is to show that systematically using local material in foreign language teaching will motivate students and develop multiculturalism as well as a readiness for intercultural communication, and this will in turn increase motivation, since the students will gain knowledge of local history and the culture of the Kazakh people. The implementation of local materials in English teaching for non-linguistic majors is accomplished by using various texts of ethno-cultural content. Having mastered local features, it is important to achieve a high level of development of a foreign language and sufficient multicultural competence. It is important to see the student as a person, taking into account his/her age, characteristics, interests, and level of speaking abilities.

Ultimately, this study will seek to answer the following question:

Is there a significant difference in competency in the foreign language between the learners who are taught in accordance with the local material and those who learn the target language solely based on the target culture?

## 2. Literature Review

In connection with the radical changes taking place in political, economic, and spiritual life, our republic needs specialists who are able to think creatively, independently solve problems that arise, build relationships, and speak several languages.

Today's socioeconomic transformations require a person to be highly professional, which is inextricably linked with the creation of conditions for people to fulfill their worth and to assert themselves. Researchers have shared their ideas about the importance of local materials in EFL teaching.

For instance, according to Aubakirova K.F. the location-based component of training should reflect social experience, the experience of an emotional value relationship, the methods of activity and creativity, and are necessary for a future specialist [5].

Local materials as the core of a cultural component need to be integrated in language instructions [6]. It is widely believed that language and culture are closely related since both elements intermingle to create manners of perceiving and expressing about the world each community lives in. The cultural component has always been one of the most essential dimensions of intercultural language teaching and is based on the following principles: awareness of one's own culture; study of the target-language culture; and comparison of the cultures under consideration. Thus, the process of learning more about the interrelationship between culture and language within the native environment led the way to consider the effect of the second culture on learning the second language [7].

The Law of Education in Kazakhstan mandates local cultural-based education. Local cultural values should be considered in efforts to improve the teaching and learning process.

Integrating local culture into the curriculum and teaching and learning process promotes the development of a creative personality, allowing students to select cultural values from which to write a report about their daily lives [8]. The cultural component of education is understood as the fostering of an ethnocultural personality by means of a foreign language. In terms of English language, local materials may be understood as a manifestation of the government's vision and one of the components of local content that can also be taught in high schools.

The cultural component in the pedagogical process should be based on modern concepts of ethnicity, represent a special sociocultural system, strive to familiarize the individual with national and world culture, and create conditions for entering into world culture. It must be considered as one of the main characteristics of education, which includes:

- a set of ideas, views, theories of socialization, development, upbringing, and education of the individual, based on the national cultural paradigm and determining the educational policy of the state;
- the process and result of the education and training of the student as a subject of the nation or ethnic group;
- the system of continuity of cultures and satisfaction of cultural needs and personality;
- the cultural tradition and language;
- the system of people's ideals, values, and moral standards;
- the level of development of modern culture [9].

This characterization defines the cultural orientation of education and the principle of the orientation of the individual as a citizen of a multi-ethnic state capable of self-determination under the conditions of modern civilization. It follows that the value of education takes into account the interests of the individual as an ethnic group and as a citizen of Kazakhstan and the world.

Since this study is more focused on local materials as a means of improving motivation, some researchers consider motivation in different ways. In the manuscript by Zoltan Dörnyei, the portion "Cognitive components of motivation" is especially worthy of this study [10]. Peter MacIntyre outlined the different versions of motivation strategies in language learning [11]. Both parts of motivational are appropriate for this study because local materials set the direction of motivation research in language learning.

Another interesting view was provided by Suresh Canagarajah (1993) who initiated discussion on issues with similarities to this study about students' opposition and willingness to engage with the local issues [12].

In a multinational state, account must be taken of the interests of the active subject of education, namely, ethnicity. Therefore, the multi-ethnic state is one of the central problems in cultural orientation and the search for a balance between the needs and interests of the individual, ethnicity, and society. The culture of the individual is the pinnacle that focuses all aspects and levels of human and group cultures. National culture, as a culture of one of the most stable human communities, is a significant factor in the development of the individual. Hence there is a need to select the components of the national culture and to organize its systematic inclusion in the educational process of the institution.

The individual, as a subject of culture, is a multi-level, complex educational component encompassing ethnic, national, educational, confessional, and other aspects, focusing on all aspects and levels of human and group cultures. However, the basic level of personal culture is the level of national culture. Development of the content of training in this area should be systematic and include components such as purpose, principles, content, methods, tools, and forms.

### 3. Local Materials Introduction in Language Teaching

The need to study local materials in the process of training specialists is the primary task of educational policy, and it is generally recognized in foreign language classes. However, teaching practice suggests that the study of local materials remains at the level of declaration. However, teaching practice suggests that the use of local materials remain on

the level of self task of students. The use of geographic-based cultural/regional material in foreign language teaching is dictated by the specifics of the language as a means of foreign language communication and has been observed throughout the long history of foreign language teaching in different countries. There is a need to learn the language through the culture of the country of the language being studied. In this case, it is important for the student to compare his/her sociocultural experience with the sociocultural experience of the native speakers.

Furthermore, local materials promote language acquisition while also broadening students' language awareness because they provide a particularly effective means of promoting language acquisition within relevant and memorable situations for processing and comprehending a new language. In addition, publications with similar substance are appropriate instructional material for a variety of exercises aimed at improving students' foreign language skills. Local materials increase motivation, which is essential for language acquisition success [13].

*Structural Content and Functional Model of Local Materials for Increasing Motivation in Learning a Foreign Language*

The systematization and design of texts with local materials and their introduction into the educational process led to the development of a structurally meaningful and functional model for increasing motivation in learning a foreign language as a means of teaching English at faculties of non-linguistic specialties.

We understand the model as "... an artificially created object similar to the process of cognition" [14].

The main design or point of concentration of the content of learning is to create a new paradigm of education. The education paradigm is a methodologically sound reference for the construction of the natural and cultural model of the school as a center of intellectual reference knowledge, high morality, and stable development in accordance with the State or international standards in the context of the new civilizational approach.

The creation of any model requires clarification of methodological approaches that contribute to increasing the importance of local materials and allows us to connect the past and present, fill the information gap, and connect national processes with local/regional ones, highlighting their main features, and helping to understand development prospects. Studying and comprehending the region enables a better understanding of those plans that must be implemented in the conditions of improving modern society while preserving the national and cultural heritage of the fatherland.

In the framework of the indicated methodological approaches, local materials are considered as the most important component of culture, the main channel for its transmission to the younger generation, a means of accumulating knowledge and developing the necessary skills for future specialists and provide for the achievement of a higher goal—the establishment of the personality of a growing person as a carrier and creator of culture.

The didactic basis on which our model is based are the following principles of content selection: the principle of regionality, the principle of systematic and subsequent relevance, the thematic principle, the principle of technology diffusion, the principle of cognition, the principle of tolerance, the principle of comparability, as well as the principle of individuality.

According to the principle of regionality, it involves the use of local history material in close connection with other subjects of the university program (history, cultural disciplines, etc.).

According to Galskova N.D., the principle of systematicity and consistency involves the assimilation of knowledge and the formation of skills and abilities in a certain logical connection, which represent a holistic education and system. The principle of systematicity and consistency is determined by the purpose of the methodology of teaching foreign languages and the laws of mental and physical development of students and is implemented when compiling programs and textbooks in a foreign language under the guidance of the teacher and the cognitive activity of students in its requirements for them [15].

The thematic principle is aimed at developing thematic material for each lesson in accordance with a standard textbook or at creating new copyrighted curricula and manuals.

The thematic principle is relevant to justify our proposed model of teaching a foreign language through regional material, not only because, according to the thematic principle, all material is divided into topics and subtopics, but also because the technique of "passing through" this material is based on the cyclical organization of the educational process. Each of the foreseen cycles is designed for a certain number of hours. A separate cycle is considered as a complete independent period of study aimed at solving a specific problem in achieving the common goal of mastering a foreign language [15].

The principle of technologization regulates the selection of teaching methods and techniques for specific regional material in the study of each topic and the use of gaming technologies in the process of teaching a foreign language using regional material.

The principle of technologization is closely related to a principle of classical didactics such as visualization, since, with the help of vision and sound means that events and facts can be presented more intelligibly and clearly [16].

The model we are developing is based on visual-audio equipment, in which a geographical map of the Zhezkazgan region, made by the students themselves, contains geographical names, sets of illustrations and postcards, slideshows on specific topics, books, magazines, newspapers, various models, and souvenirs.

This connection is of particular importance as using local-history material, requires students to have a certain theoretical knowledge in various fields and perform active cognitive work which has the greatest effect when students themselves participate in their development and construction.

This principle occupies a special place in our models since relying on it creates the conditions for sensory perception and introduces the native reality for students into the educational process. Mastering a foreign language is impossible without using this principle. The formation of the abilities of a coherent statement and conducting a conversation in the language being studied is unthinkable without the widespread use of visibility, which allows you to simulate a communication situation and stimulate monologic and dialogical speech.

The principle of cognition provides students with the knowledge of cultural/regional-based materials, new information, and new knowledge in the process of foreign language teaching.

The intensification of the cognitive activity of students, the development of interest in the subject, the formation of independence, and a creative attitude towards the student will, in our opinion, be much more successful if the principle of cognition is observed.

The principle of tolerance is aimed at understanding the cultural condition of communication and human behavior, overcoming ethnic and cultural centrism and ethnic and cultural bias and xenophobia, which should ultimately contribute to the formation of multicultural tolerance in the field of everyday intercultural interaction [17].

The principles of comparability and integrativity regulate the use of culture-based materials along with the materials of a region's geographic character and ensure the "convocation" of languages and cultures. There is no doubt of the need to study the interconnection of languages and cultures since, in the framework of one's own culture, an illusion is created that their own world, worldview, and lifestyle is the only one possible, and this is absolutely unfair. The principle of comparativism and the reliance on it in the process of teaching foreign languages with the use of local history tools, allow us to hope to overcome both the language barrier and the cultural barrier, sometimes causing a cultural shock. Ter-Minasova S.G. (2003) rightly noted that all the subtleties and the whole depth of the problems of inter-linguistic and intercultural communication become especially visual, and sometimes simply realized, when comparing foreign languages with relatives and a foreign culture with a native. In addition, this perception of a foreign culture through the prism of its own culture, through a vision of its own history, and through a love for the native land, will give a positive result and will lead to real success both in preparation for

intercultural communication and in it itself [17]. Wilhelm von Humboldt argued, "Through the diversity of languages, the wealth of the world and the diversity of what we know in it are revealed to us; and human being becomes wider for us, because languages in distinct and effective terms give us different ways of thinking and perception".

We also note that when using local materials, it is necessary to adhere to an individual approach, trying to more fully take into account the interests and inclinations of students. In the process of teaching a foreign language, it is necessary to ensure the simultaneous formation of a general and special personality and individuality in a person which is inherent in this student—a representative of a particular ethnic group, territory, state, and humanity as a whole. An individual approach requires a deep study of the students' inner world, the level of their knowledge, good breeding, the conditions in which training and education are conducted, and the formation of their personality [18].

All these principles are interconnected and implemented in unity. Their use in the process of teaching a foreign language gives a humanistic orientation to the learning process, making it effective and efficient. It seems to us that the principles that have been mentioned and analyzed by us have reflected the most important laws of training and education.

The selection of language material is dictated by the limited possibilities of memorization and the number of hours that are devoted to learning a foreign language in a non-linguistic university. Therefore, each language unit and each acquired skill must find application in subsequent work on the language. At the same time, language material that is not used to realize the ultimate goals of learning should not be given simply "for future use" or for the convenience of developing skills and other non-essential factors [19].

When selecting material, it should be borne in mind that all linguistic minima are closely connected with each other, but the leading component is the dictionary. It is selected first of all, because a set of vocabulary determines the grammatical minimum standard.

Based on the analysis of the principles of selecting vocabulary for the purposes of this study, the following were put forward as leading candidates: the thematic principle, the principle of communicative significance, the principle of taking into account the content of textbooks, and the principle of taking into account the national cultural identity. The implementation of the thematic principle in the selection of lexical units with a cultural component is as follows: the selected vocabulary with a cultural component should be relevant to the topic and include the most important concepts that reflect the specifics of the culture of native speakers and native culture within these topics. The selection procedure in accordance with this principle consists in dividing the topic into subtopics in which concepts that reflect the region and local historical reality are highlighted. Further, to express this information, a selection of vocabulary in compliance with the principle of communicative significance consists in a reasonable selection of the required number of lexical units to ensure the need for verbal communication to describe the native culture of students in the framework of topics defined by the university and school curriculum for this stage of education.

## 4. Methods

An experimental method was conducted in order to identify the level of development of educational motivation in the course of using local materials and check the effectiveness of our structural model of local materials use in EFL classes. The experiment consisted of three stages.

The aim of the first stage was to identify the initial level of motivation of students to learning a foreign language.

The aim of the second stage was integrating local materials into the learning process.

The aim of the third stage was to compare the level of students' motivation after the experiment with their initial level.

In order to indicate the motivation level of students to learn a foreign language we have set the following levels (see Table 1).

**Table 1.** The levels of motivation to learning a foreign language learning.

| Level | Meaning | Indicators |
|---|---|---|
| Optimal level | Strong motivation to learn a foreign language | 71–100% |
| Acceptable level | Seak motivation to learn a foreign language | 41–70% |
| Critical level | Unmotivated to learn | 1–40% |

## 5. Participants

The participants of the research were equally assigned into the experimental group and the control group. They were students majoring in pedagogical specialties from L.N.Gumilyov Eurasian National University in Nur—Sultan city. The participants of this study comprised B1-level students who were at the intermediate level at the time of the study. The experimental group's average age was 18 years and the control group's average age was 18–19 years. The participating students were all native speakers of Kazakh and none of them had stayed in English-speaking countries for more than a week.

## 6. Procedure

At the first stage of the experimental work, we used the questionnaire method to identify the initial level of educational motivation to study local materials.

The questionnaire consisted of 18 closed questions measured using a Likert scale (very important, important, and not important). Answers reflected students' attitudes to the importance of using local materials in the English learning process.

The questionnaire included personal information about students, their level of English proficiency, what part of Kazakhstan they are from, reasons for learning English, what materials they would like to use to learn English, personal opinions related to local materials, and their attitude to learning local culture and history.

At the second stage, the structural-functional model's content was integrated into EFL classes. Table 2 illustrates the content of local materials.

**Table 2.** Structural-functional model of local materials for increasing motivation to learning a foreign language.

| Education Material Structure | | Purpose and Functions | | |
|---|---|---|---|---|
| | | Cognitive | | Educating |
| | | Language knowledge | Local knowledge | |
| Program-based material | Local material | Language knowledge | Local knowledge | Formation of a respectful humane attitude to the culture heritage of the past; self- awareness as a citizen of the Republic of Kazakhstan |
| Great Britain. Physical Features | Ulytau | Acquisition of communicative competence (CC); thematic principle of vocabulary selection; activation of the grammatical minimum; approximate pronunciation training. | The historical role of Ulytau in the formation of Kazakh statehood; The horizons of students expands and their cognitive interest develops. | |
| The USA. Government in America: People, Policy. Cities of the USA | The mausoleum of Alasha-khan | Acquisition of CC; the principle of communicative significance in the selection of vocabulary; activation of the grammatical minimum; approximate pronunciation training. | Getting information about one of the great lords Alasha Khan; the development of speech is accompanied by positive emotions (a sense of pride, interest). | The formation of multiculturalism; promotes patriotic education, general political education |

**Table 2.** *Cont.*

| Education Material Structure | | Purpose and Functions | | |
| --- | --- | --- | --- | --- |
| | | Cognitive | | Educating |
| | | | Cognitive | Educating |
| Places of interest in Great Britain | The mausoleum of Zhoshy- khan | Acquisition of CC; the principle of accounting for the content of textbooks in the selection of vocabulary; activation of the grammatical minimum; approximate pronunciation training. | Getting information about one of the sons of Genghis Khan; assimilation of individual linguistic phenomena and linguistic and geographical realities in their comparison. | Promotes a better understanding of foreign language culture and the formation of increased tolerance to the participants in communication. |
| Places of interest in the USA | The mausoleum of Dombaul | Acquisition of CC; thematic principle of vocabulary selection; activation of the grammatical minimum; approximate pronunciation training. | Getting information about Dombaul according to various legends; comparison and juxtaposition of the realities of the cultural and everyday life of the country of the studied language and Kazakhstan. | The formation of national self-awareness and the development of an emotional-value attitude of an individual to the culture of his people and their history. |
| Customs and Traditions in the USA | Rock drawings Of "Terekty-Aulie" | Acquisition of CC; the principle of taking into account the national identity of culture in the selection of vocabulary; activation of the grammatical minimum; approximate pronunciation training. | Acquaintance with one of the most interesting places of ancient rock art, the components of which are petroglyphs. | Raises the ideological level of students and the formation of moral beliefs and skills |
| Customs and Traditions in Britain | Summary of the Zhezkazgan Region's Monuments | Acquisition of CC; thematic principle of vocabulary selection; activation of the grammatical minimum; approximate pronunciation training. | Gaining knowledge of new archaeological data on monuments in the vast Sary-Arka; contribute to greater readiness for intercultural communication | They have a significant impact on the formation of the worldview of students and on the integration of scientific knowledge. |

At the third stage, questionnaires identical to those used in the first stage were given to the students in order to determine whether the local materials could increase students' motivation levels.

## 7. Results

The following results were revealed before and after the experiment.

According to the results of the conducted questionnaire we determined the following motivation levels of students in both groups before the experiment. This is shown in the Tables 3 and 4:

**Table 3.** The initial level of motivation of students in both groups.

| Groups | Indicator | Levels | | | Total |
| --- | --- | --- | --- | --- | --- |
| | | Critical | Acceptable | Optimal | |
| EG | Quantitative | 15 | 16 | 1 | 32 |
| | % | 46.88% | 50% | 3.12% | 100% |
| CG | Quantitative | 10 | 21 | 1 | 32 |
| | % | 31.25% | 65.63% | 3.12% | 100% |

**Table 4.** Dynamics of changes in indicators of the development of motivation to learn a foreign language (beginning and end of experiment).

| Groups | Indicator | Period | Levels | | | Total |
|---|---|---|---|---|---|---|
| | | | **Critical** | **Acceptable** | **Optimal** | |
| EG | Quantitative % | Beginning | 15 46.88% | 16 50% | 1 3.12% | 32 100% |
| | Quantitative % | End | 6 18.76% | 23 71.88% | 3 9.36% | 32 100% |
| CG | Quantitative % | Beginning | 10 31.25% | 21 65.63% | 1 3.12% | 32 100% |
| | Quantitative % | End | 9 28.13% | 22 68.75% | 1 3.12% | 32 100% |

Student's *t*-test was used in order to determine the correlation of the independent samples at the beginning and end of the experiment. (See Table 5).

**Table 5.** Independent Samples Test.

| | | Levene's Test for Equality of Variances | | *t*-Test for Equality of Means | | | | | | |
|---|---|---|---|---|---|---|---|---|---|---|
| | | **F** | **Sig.** | **t** | **df** | **Sig. (2-tailed)** | **Mean Difference** | **Std. Error Difference** | **95% Confidence Interval of the Difference** | |
| | | | | | | | | | **Lower** | **Upper** |
| EG | Equal variances assumed | 0.709 | 0.403 | −0.503 | 62 | 0.617 | −1.063 | 2.113 | −5.286 | 3.161 |
| | Equal variances not assumed | | | −0.503 | 61.972 | 0.617 | −1.063 | 2.113 | −5.286 | 3.161 |
| CG | Equal variances assumed | 0.170 | 0.682 | 3.828 | 62 | 0.000 | 7.188 | 1.878 | 3.434 | 10.941 |
| | Equal variances not assumed | | | 3.828 | 61.857 | 0.000 | 7.188 | 1.878 | 3.434 | 10.941 |

As we can see from the table, $t = 3.8$ when $p < 0.001$. This means that our model influenced the learning process and students were motivated through local materials.

## 8. Discussion

The current study focused on the impact of including local materials, as well as participant preferences and opinions regarding the significance of local materials in language learning. This study aimed to make a contribution to the field of EFL and fill the gap related to increasing motivation through local materials. This paper's major goal was to raise awareness of how the integration of local materials affects students learning English in terms of their competency levels.

This study added to the ongoing debate of whether or not to include local materials in classes where students are learning foreign languages. The results imply that teaching English while including local materials positively impacts learning outcomes and significantly motivates students.

It might be claimed that the findings are consistent with those of past studies on the integration of local materials and the target language. According to the results of the research, courses that incorporate aspects of the local materials are more engaging and appealing to the target audience, and students would have more in common with native speakers while learning a foreign language.

The inclusion of local content in the curriculum helps students overcome their inability to communicate their cultural and national values in English.

The notion that using local content to teach English is one of the most effective methods for fostering local culture was backed by Kaltsum and others. Systematized and well-selected resources will enhance teaching methods and produce favorable outcomes [20].

Eldin believed that exceptional characteristics of local materials have a significant impact on language learning [21]. One believes that language and culture are instruments for realizing the concept of communicative direction. It is an important component of language learning that promotes comprehension and progress. Qu suggested using a variety of resources to educate culture, highlighting the value of conversations, mini-dramas, role-playing, dances, songs, photographs, films, and bulletin boards [22]. Almujaiwel (2018) argued that increasing students' familiarity with local cultures improves their language skills and their ability to interact with foreigners in English [23].

Thus, materials of local content are an integral part of the general system of educational work in a foreign language, play a significant role in improving the quality of education, and are one of the most effective means of maintaining motivation to study a subject.

As a result of this pedagogical experiment, we discovered that students' educational motivation to study EFL has increased. The results of the trial validated the research hypothesis: using materials with local relevance in EFL classes will improve motivation to learn a foreign language. Local materials, with their enormous spiritual potential, served as a source of inspiration for educational and cognitive activity, enhancing the educational process and making it more personal for each student.

**Author Contributions:** Conceptualization, G.T. and D.K.; Formal analysis, J.S.; Investigation, G.T. and D.K.; Methodology, K.K.; Project administration, J.S.; Supervision, K.K.; Visualization, N.K. All authors have read and agreed to the published version of the manuscript.

**Funding:** This research received no external funding.

**Institutional Review Board Statement:** This study was conducted in accordance with the Declaration of Helsinki and approved by the Institutional Review Board of Eurasian National University.

**Informed Consent Statement:** Informed consent was obtained from all people involved in the study.

**Data Availability Statement:** Data was obtained through https://docs.google.com/forms/d/1-jz0 K2rH_Za-8BRkkslaYR5uVm6698yXqL4JSNref40/edit.

**Conflicts of Interest:** The authors declare no conflict of interest.

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
