# Peer review of "Local Materials as a Means of Improving Motivation to EFL Learning in Kazakhstan Universities"

_education, doi:10.3390/educsci12090604_

Round 1
Reviewer 1 Report
Minor spell check of English required.
Author Response
Dear Reviewer,
We would like to thank you for your interest in our topic and appreciate your review.
Reviewer 2 Report
The article deals with the rarely discussed topic of the motivation of non-philological students to learn English in Kazakhstan. The choice of topic is accurate and the goals formulated by the authors are substantive: the necessity for focused usage of local content resources in foreign language study. However, the article requires considerable changes, especially in terms of the literature review. Almost all of the sources are in Russian, with no leading authorities on motivation such as Zoltan Dörnyei, Peter MacIntyre or Howard Gardner.
In introduction the authors write:
“This topic has been the subject of numerous writings by teachers, psychologists, and other experts”. Here some names should be provided.. There are also no references to earlier research on the selected topic. The authors are definitely not the first to become interested in this subject. This is then helpful in Discussion.
I would also suggest more caution in formulating conclusions. The statement: “ It is concluded that by providing local materials the instruction will be sufficient to assure student success” is an exaggeration. If success in learning a language could be guaranteed in such a simple way, it would probably be long described in literature.
The study description should follow the pattern:
1. Hypotheses/ research questions (missing)
2. Participants (missing)
3. Method/procedure
4. Results and interpretation
“The program-based materials do not require comment, because they are based on a training program in accordance with the requirements of the State Educational Standard of the Republic of Kazakhstan”. Well, I think a word of comment would be helpful (sources of materials, level, etc.)
The section “Discussion” traditionally includes a discussion about received results and some comparisons/ contrasts with previously published research and not theory.
A section/ sections “conclusion/ study limitations/ pedagogical” implications are required in academic papers.
The language is overall acceptable, however I suggest a thorough proofreading, e.g. P7 “In order identify” section “discussions” (articles, conjunctions).
Also short statements (in abstract) “The result of this study confirms that familiar material will help students in learning a foreign language. This study's findings support the idea that recognizable content might contribute successful learning a foreign language. In general, motivation plays an important influence in the rate and success of foreign language learning” should be polished to make the style more academic.
Author Response
Dear Reviewer,
We would like to thank you for your review and take into consideration all the comments you suggested us.
The article deals with the rarely discussed topic of the motivation of non-philological students to learn English in Kazakhstan. The choice of topic is accurate and the goals formulated by the authors are substantive: the necessity for focused usage of local content resources in foreign language study. However, the article requires considerable changes, especially in terms of the literature review. Almost all of the sources are in Russian, with no leading authorities on motivation such as Zoltan Dörnyei, Peter MacIntyre or Howard Gardner.
Thank you very much for prompts. We have studied proposed papers and added them into the literature review.
In introduction the authors write:
“This topic has been the subject of numerous writings by teachers, psychologists, and other experts”. Here some names should be provided.. There are also no references to earlier research on the selected topic. The authors are definitely not the first to become interested in this subject. This is then helpful in Discussion.
We have added some names who did earlier research in our topic.
I would also suggest more caution in formulating conclusions. The statement: “ It is concluded that by providing local materials the instruction will be sufficient to assure student success” is an exaggeration. If success in learning a language could be guaranteed in such a simple way, it would probably be long described in literature.
The study description should follow the pattern:
- Hypotheses/ research questions (missing)
- Participants (missing)
- Method/procedure
- Results and interpretation
We have revised the paper with the suggested structure. Research questions have been added into the introduction section. The information about the method, participants, procedure and discussion have been added.
“The program-based materials do not require comment, because they are based on a training program in accordance with the requirements of the State Educational Standard of the Republic of Kazakhstan”. Well, I think a word of comment would be helpful (sources of materials, level, etc.)
We found it useless and deleted it.
The section “Discussion” traditionally includes a discussion about received results and some comparisons/ contrasts with previously published research and not theory.
A section/ sections “conclusion/ study limitations/ pedagogical” implications are required in academic papers.
Discussions and conclusions have been rewritten.
The language is overall acceptable, however I suggest a thorough proofreading, e.g. P7 “In order identify” section “discussions” (articles, conjunctions).
Colleague from US university helped with editing
Also short statements (in abstract) “The result of this study confirms that familiar material will help students in learning a foreign language. This study's findings support the idea that recognizable content might contribute successful learning a foreign language. In general, motivation plays an important influence in the rate and success of foreign language learning” should be polished to make the style more academic.
Done.
Thank you again for your understanding and support.
Reviewer 3 Report
I liked the choice of the topic and found it interesting. Also, the framework is helpful and could be used in other research projects.
The greatest problem that prevents the reader from appreciating and benefitting from the text is the obscurity of the style. The manuscript has been prepared in a rather careless manner with occasional enevenness of the style. A good proofreading as far as the style is concerned would make the article more friendly in reception. Also, the direct addressing of the reader in the form of the imperative (e.g., "Consider", p. 6) seems inadequate.
As to the content, I don't think "personality-focused" issues (p. 2) should be dealt with in the text, as it is a completely another aspect, to my mind, too loosely related to the issue of local materials.
The same refers to the abstract which contains repetitions (some sentences convey the same content, conclusions are given before results, can't see the purpose of the last sentences, etc.).
References should be enriched. I suggest considering the text by Suresh Canagarajah in TESOL Quarterly · December 1993 who initiated discussing a bit similar issues, that is students' opposition and willingness to engage with the local issues.
Author Response
Dear Reviewer,
We would like to thank you for your review and take into consideration all the comments you suggested us.
I liked the choice of the topic and found it interesting. Also, the framework is helpful and could be used in other research projects.
The greatest problem that prevents the reader from appreciating and benefitting from the text is the obscurity of the style. The manuscript has been prepared in a rather careless manner with occasional enevenness of the style. A good proofreading as far as the style is concerned would make the article more friendly in reception. Also, the direct addressing of the reader in the form of the imperative (e.g., "Consider", p. 6) seems inadequate.
Colleague from US university helped with editing. imperative (e.g., "Consider", p. 6) seems inadequate. – changed.
As to the content, I don't think "personality-focused" issues (p. 2) should be dealt with in the text, as it is a completely another aspect, to my mind, too loosely related to the issue of local materials.
This part has been taken out.
The same refers to the abstract which contains repetitions (some sentences convey the same content, conclusions are given before results, can't see the purpose of the last sentences, etc.).
The abstract has been rewritten.
References should be enriched. I suggest considering the text by Suresh Canagarajah in TESOL Quarterly · December 1993 who initiated discussing a bit similar issues, that is students' opposition and willingness to engage with the local issues.
Has been studied and added.
Round 2
Reviewer 2 Report
no comments